# Hyperbolic Multimodal Representation Learning for Biological Taxonomies

**ZeMing Gong**[1]     **Chuanqi Tang**[1]     **Xiaoliang Huo**[1]     **Nicholas Pellegrino**[2]
**Austin T. Wang**[1]     **Graham W. Taylor**[3,4]     **Angel X. Chang**[1,5]
**Scott C. Lowe**[3†]     **Joakim Bruslund Haurum**[6†]

Simon Fraser University[1]     University of Waterloo[2]     Vector Institute[3]
University of Guelph[4]     Alberta Machine Intelligence Institute (Amii)[5]     Aalborg University[6]

{zmgong, cta156, xiaoliang_huo, atw7, angelx}@sfu.ca, nicholas.pellegrino@uwaterloo.ca,
gwtaylor@uoguelph.ca scott.lowe@vectorinstitute.ai, joha@create.aau.dk

## Abstract

*Taxonomic classification in biodiversity research involves organizing biological specimens into structured hierarchies based on evidence, which can come from multiple modalities such as images and genetic information. We investigate whether hyperbolic networks can provide a better embedding space for such hierarchical models. Our method embeds multimodal inputs into a shared hyperbolic space using contrastive and a novel stacked entailment-based objective. Experiments on the BIOSCAN-1M dataset show that hyperbolic embedding achieves competitive performance with Euclidean baselines, and outperforms all other models on unseen species classification using DNA barcodes. However, fine-grained classification and open-world generalization remain challenging. Our framework offers a structure-aware foundation for biodiversity modelling, with potential applications to species discovery, ecological monitoring, and conservation efforts.*

## 1. Introduction

Specimen identification is an essential step for monitoring and mitigating biodiversity loss, requiring accurate classification of organisms within the taxonomic hierarchy across diverse ecosystems. DNA barcodes [2, 10] provide a way to classify specimens to known taxa or identify them as novel to science, but classification to the species level remains challenging, especially when barcodes are unavailable. To tackle this, CLIBD [8] showed that using contrastive learning to align DNA barcode embeddings to image embeddings can improve classification at the species level even when restricted to only using images for inference.

However, a key limitation of CLIBD [8] and other recent biodiversity-focused multimodal methods [19] is that they do not utilize the known taxonomic hierarchy of the

---

†Equal advising.

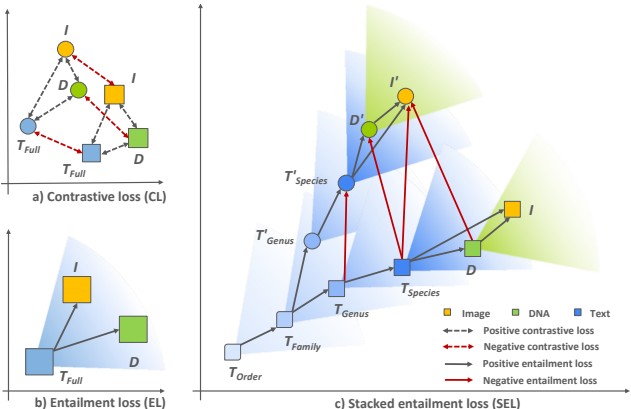

Figure 1. (a) **Contrastive loss**: instance-level alignment between modalities. (b) **Entailment loss**: enforces hierarchy within the text modality using entailment cones. (c) **Stacked entailment loss**: combines EL and cross-modal constraints by aligning image and DNA embeddings to multiple levels of the text hierarchy.

input data. Motivated by the effectiveness of hyperbolic embeddings for capturing hierarchical relationships [3], we explore whether embeddings in hyperbolic space can provide more accurate fine-grained classification. Our model takes inputs from multiple modalities—DNA barcodes, specimen images, and hierarchical taxonomic labels—and is trained to co-align their embeddings into a shared hyperbolic space to promote taxonomic alignment across modalities.

Our experimental results show that our hyperbolic multimodal learning framework achieves strong performance in taxonomic classification and retrieval, especially at higher taxonomic ranks. The approach consistently matches or outperforms Euclidean baselines. However, all methods—including ours—face challenges in fine-grained species classification, particularly for previously unseen taxa. These results highlight both the potential of hyperbolic learning for hierarchical biological data, and the ongoing difficulty of open-world classification for biodiversity.

## 2. Related Work

**Euclidean Multimodal Learning** is the norm for recent advances in the multimodal contrastive learning domain, both in general vision-language frameworks such as CLIP [17] and SigLIP [22] and domain-specific ones, including those for biodiversity applications [8, 9, 19]. These biodiversity models embed images, textual data, and optionally DNA barcodes into a shared Euclidean embedding space using modality-specific encoders and contrastive learning objectives. CLIBD [8] in particular demonstrates zero-shot classification on BIOSCAN-1M [7], achieving superior accuracy to unimodal baselines.

**Hyperbolic Representation Learning** is an approach that utilizes hyperbolic geometry to encode features into a hierarchical representation space [14]. Unlike Euclidean space, hyperbolic spaces grow exponentially, matching the way the number of nodes in a hierarchy can grow exponentially with the depth. Nickel and Kiela [15] showed that taxonomic relationships in language can be effectively captured using hyperbolic embeddings. Recently, hyperbolic visual representation learning has been applied to vision tasks such as image retrieval [11] and image segmentation [6]. While the majority of these works use hyperbolic geometry only at the last layer, recent advances have been made towards developing *fully* hyperbolic models, *e.g.*, Poincaré ResNet [21].

**Hyperbolic Multimodal Learning** combines multimodal learning and the use of hyperbolic geometry to co-align embeddings from different modalities in a hierarchical representation space. Liu et al. [13] showed how to align images and text embeddings in a Poincaré hyperbolic space, while MERU [3] uses contrastive learning to align images and text in Lorentzian space. Following MERU, HyCoCLIP [16] incorporated compositional constraints to strengthen fine-grained alignment between parts and wholes in visual concepts. These works show hyperbolic geometry can enhance the structural consistency and interpretability of multimodal models, particularly in settings with implicit or weakly defined hierarchies.

Our method differs from prior work in three key ways. First, rather than focusing on vision-language, we incorporate biologically grounded modalities—DNA barcodes and taxonomic labels—that are more salient for species-level classification. Second, we leverage *explicit* taxonomic hierarchies to guide representation learning rather than relying on *implicit* hierarchical signals such as caption specificity or object part composition. Third, our stacked entailment loss enforces consistency across hierarchical ranks.

## 3. Approach

We propose a multimodal representation learning framework that unifies specimen DNA barcodes, images, and taxonomic labels into a shared *hyperbolic* embedding space.

By leveraging hyperbolic geometry, we aim to preserve hierarchical taxonomic relationships, improving classification accuracy and representation quality across the hierarchy.

Our framework employs three specialized encoders to process each of the data modalities: an image encoder extracts visual features, a DNA encoder encodes genetic sequences, and a text encoder captures semantic information from taxonomic labels of varying depth. These encoders independently map their inputs into a common embedding space, in which contrastive learning aligns multimodal representations for downstream tasks. We expand on CLIBD [8] by lifting the embeddings into hyperbolic space, and evaluate on the BIOSCAN-1M dataset [7].

### 3.1. Input and Output Specification

During training, we jointly optimize the encoders using triplets of aligned data—specimen image, DNA barcode, and hierarchical taxonomic labels (*e.g.*, "Order: Diptera; Family: Syrphidae; Genus: Episyrphus; Species: Episyrphus balteatus")—so that their embeddings are both cross-modally aligned and geometrically consistent with the taxonomic hierarchy. This objective supports flexible inference with any subset of modalities while preserving multi-level taxonomic relationships in the learned space. At inference time, the model supports both uni- and cross-modal retrieval, allowing it to taxonomically classify specimens using any available combination of images, DNA barcodes and taxonomic labels. This enables robust downstream use in biodiversity monitoring and taxonomic classification, even with missing or noisy modalities.

### 3.2. Encoders

We adapt the experimental setup from Gong et al. [8], using pretrained ViT-B/16, BERT-Small, and BarcodeBERT encoders for image, text, and DNA barcode modalities.

Each encoder produces Euclidean embeddings, which are then projected into a Lorentzian hyperbolic space with curvature $c$, via an exponential mapping centred at the origin. We refer the reader to Desai et al. [3] for details. The shared space enables contrastive alignment across modalities while preserving the hierarchical taxonomic structure.

### 3.3. Stacked Entailment Loss

To better leverage the inherent structure of the biological taxonomy, we propose a hierarchical learning objective termed *stacked entailment loss* (SEL). This mechanism is designed to explicitly enforce geometric relationships between taxonomic ranks—order, family, genus, and species—within hyperbolic space (see Figure 1). The design is inspired by compositional entailment mechanisms introduced in prior work [16], but adapted to reflect the nested and non-overlapping nature of biological hierarchies.

The core idea is to constrain the embeddings of lower-level taxa (*e.g.*, genus) to lie within an entailment cone of their parent nodes (*e.g.*, family). This entailment constraint is applied between each consecutive pair of levels in the hierarchy to ensure each child node is within the space "above" its parent, using a margin-based loss. Additionally, we introduce a *negative entailment* loss term which ensures each child node is *not* within the space "above" nodes from the preceding layer that are *not* its parent.

Given a batch $\mathcal{B} = \{(x_i, y_i, c_i)\}_{i=1}^B$, where $x_i$ and $y_i$ are embeddings and $c_i$ the class, we define positive pairs $\mathcal{P} = \{(i,j) : c_i = c_j\}$ and negative pairs $\mathcal{N} = \{(i,j) : c_i \neq c_j\}$. The corresponding entailment losses are:

$$L_{\text{ent}}^+ = \frac{1}{|\mathcal{P}|} \sum_{(i,j) \in \mathcal{P}} \max\left(0, \, \text{ext}(x_i, y_j) - \text{aper}(x_i)\right) \quad (1)$$

$$L_{\text{ent}}^- = \frac{1}{|\mathcal{N}|} \sum_{(i,j) \in \mathcal{N}} \max\left(0, \, \text{aper}(x_i) - \text{ext}(x_i, y_j) + m\right) \quad (2)$$

where $\text{ext}(x, y)$ denotes the exterior angle between $x$ and $y$ in hyperbolic space, $\text{aper}(x)$ is the cone aperture of $x$, and $m$ is the margin for negative pairs. The positive and negative entailment loss are then combined into: $L_{\text{ent}} = \frac{1}{2}\left(L_{\text{ent}}^+ + L_{\text{ent}}^-\right)$. Unlike flat contrastive objectives, which treat all positive pairs equally, the stacked entailment loss introduces a directional notion of containment in the taxonomic hierarchy (from parent to child), ensuring that more specific taxa (fine-grained nodes) are properly nested under their broader ancestors in the hyperbolic hierarchy. The overall stacked-entailment loss consists of two parts: $L_{\text{SEL}} = L_{\text{SEL-intra}} + L_{\text{SEL-inter}}$. The first component, **intra-modal entailment loss**, enforces hierarchy among taxonomic labels. Let the taxonomy have $R$ levels (*e.g.*, order, family, genus, species), indexed $r = 1, 2, \ldots, R$ from root to leaf. $T_r$ is the embedding at rank $r$, and $\mathbb{1}_r$ an indicator function for the availability of the label at rank $r$. Then we construct the intra-modal stacked entailment loss,

$$L_{\text{SEL-intra}} = \frac{1}{\sum_{r=2}^R \mathbb{1}_r} \sum_{r=2}^R \mathbb{1}_r \times L_{\text{ent}}(T_r, T_{r-1}). \quad (3)$$

Secondly, we introduce an **inter-modal entailment loss** that bridges the taxonomic labels with other modalities:

$$L_{\text{SEL-inter}} = \frac{1}{3}\left(L_{\text{ent}}(I, T_{R'}) + L_{\text{ent}}(D, T_{R'}) + L_{\text{ent}}(I, D)\right) \quad (4)$$

where $I$ and $D$ are the embeddings of images and DNA barcodes respectively, and $T_{R'}$ refers to the deepest available taxonomic label (*i.e.*, $T_{\text{Species}}$ if species is known, $T_{\text{Genus}}$ if species isn't known but genus is, *etc.*). This term ensures that modality-specific inputs are not only aligned with the correct label, but also geometrically nested within the same hierarchical space. Since there can be multiple specimens with the same DNA barcode, and the same specimen can have different images, we consider the barcode to be more abstract than the image and also include an entailment loss term from barcode to image in the inter-modality objective.

In summary, our stacked entailment loss unifies taxonomic ordering and modality alignment, and embeds hierarchical structure into model training. This enables better generalization, especially with incomplete labels or unseen species. By explicitly modelling the hierarchical containment of taxonomic levels, our approach enables independent retrieval and prediction at any rank (*e.g.*, order, family, genus, or species), facilitating multi-level querying and evaluation directly within the learned representation. This stands in contrast to CLIBD, which produces predictions at all levels jointly We also extend the stacked entailment loss with two variants.

- **Image-DNA contrastive loss:** By adding a contrastive loss term based on the negative Lorentz distance between image and DNA embeddings, we encourage stronger cross-modal alignment and can improve the accuracy of image-to-DNA retrieval.
- **Full-text supervision:** We introduce an extra language input by concatenating taxonomic labels from all four ranks (order, family, genus, species), as is used in CLIBD. The full text is also used for contrastive alignment to the image and DNA embeddings.

## 4. Experiments and Results

We use the Euclidean-space CLIBD model [8] as a baseline, and adapt the CLIBD training pipeline to use hyperbolic-space based on the MERU framework [3]. We experimented with different combinations of loss functions, including entailment loss, stacked entailment loss, and contrastive loss. Experiments were conducted on four NVIDIA A100 GPUs (80GB VRAM each). We use a batch size of 2000 (4 × 500), except for experiments using stacked entailment, which could only fit a batch size of 1520 (4 × 380). All models were trained for 50 epochs with Adam [12]. The learning rate was scheduled using a one-cycle policy [18], ranging from $1 \times 10^{-6}$ to $5 \times 10^{-5}$. For the contrastive loss, we use a trainable temperature, initialized to 0.07.

### 4.1. Metrics and Datasets

We conduct experiments on the BIOSCAN-1M dataset [7], which provides high-quality images with paired DNA barcodes and taxonomic labels for over 1 million insect specimens. For simplicity, we train all models on CLIBD's `train_seen` split of BIOSCAN-1M (36 k samples), which ensures all samples have complete species-level labels. The CLIBD results reported in our experiments are likewise obtained by training on this same `train_seen` split, rather than using a pretrained CLIBD model. We leave expanding the experiments to the full BIOSCAN-1M training dataset to future work.

Similar to CLIBD, we evaluate classification performance across taxonomic ranks and for both seen and unseen classes, using class-averaged (macro) top-1 accuracy.

Table 1. Macro top-1 accuracy (%) comparison of different training objectives across taxonomic levels on BIOSCAN-1M. CL: contrastive loss. EL: entailment loss; SEL: stacked entailment loss; We evaluate uni- and multi-modal retrieval tasks including DNA-to-DNA, Image-to-Image, and Image-to-DNA. Accuracy is reported on both seen and unseen taxa, along with their harmonic mean (H.M.). Each method is further characterized by the configuration of entailment loss used (EL config.), whether full taxonomic text embedding is included utilized during training (Full Text), and the choice of embedding space (Euclidean: $\mathbb{R}^n$, or Lorentzian-hyperbolic: $\mathbb{H}_L^n$). All models are trained on the `train_seen` split of CLIBD and evaluated on the `test` split. **Best** results are shown in bold; second-best are underlined.

| Rank | Method | EL config. | Full Text | Space | DNA-to-DNA | | | Image-to-Image | | | Image-to-DNA | | |
|---|---|---|---|---|---|---|---|---|---|---|---|---|---|
| | | | | | Seen | Unseen | H.M. | Seen | Unseen | H.M. | Seen | Unseen | H.M. |
| Order | CLIBD | – | ✓ | $\mathbb{R}^n$ | **89.1** | 87.8 | 88.4 | **99.5** | **66.4** | **79.6** | **98.7** | **49.5** | **65.9** |
| | CL | – | ✓ | $\mathbb{H}_L^n$ | **89.1** | 85.6 | 87.3 | 98.5 | 61.2 | 75.5 | 89.1 | 47.8 | 62.2 |
| | EL+CL | Pos. | ✓ | $\mathbb{H}_L^n$ | 88.6 | 86.5 | 87.5 | 98.6 | 56.9 | 72.1 | 77.8 | 48.4 | 59.7 |
| | SEL | Pos.+Neg. | ✗ | $\mathbb{H}_L^n$ | 88.4 | **90.8** | **89.6** | 79.3 | 62.3 | 69.8 | **98.7** | 48.9 | 65.4 |
| | SEL+CL | Pos.+Neg. | ✗ | $\mathbb{H}_L^n$ | 88.7 | 86.3 | 87.5 | 99.4 | 65.9 | 79.3 | 78.6 | 48.2 | 59.7 |
| | SEL+CL | Pos.+Neg. | ✓ | $\mathbb{H}_L^n$ | 88.9 | 88.2 | 88.5 | 99.0 | 60.9 | 75.4 | 78.6 | 48.9 | 60.3 |
| Family | CLIBD | – | ✓ | $\mathbb{R}^n$ | 90.8 | 75.8 | 82.6 | **89.2** | **52.2** | **65.9** | **83.6** | **19.3** | **31.4** |
| | CL | – | ✓ | $\mathbb{H}_L^n$ | 90.3 | 76.6 | 82.9 | 83.9 | 48.5 | 61.4 | 79.6 | 18.8 | 30.4 |
| | EL+CL | Pos. | ✓ | $\mathbb{H}_L^n$ | 89.3 | 74.9 | 81.4 | 81.9 | 37.6 | 51.5 | 76.7 | 16.8 | 27.6 |
| | SEL | Pos.+Neg. | ✗ | $\mathbb{H}_L^n$ | 86.8 | **78.8** | 82.6 | 79.0 | 41.8 | 54.7 | 78.9 | 18.4 | 29.9 |
| | SEL+CL | Pos.+Neg. | ✗ | $\mathbb{H}_L^n$ | 89.0 | 76.9 | 82.5 | 79.6 | 46.6 | 58.8 | 78.7 | 17.3 | 28.4 |
| | SEL+CL | Pos.+Neg. | ✓ | $\mathbb{H}_L^n$ | **91.2** | 77.0 | **83.6** | 82.4 | 41.5 | 55.2 | 78.1 | 17.4 | 28.4 |
| Genus | CLIBD | – | ✓ | $\mathbb{R}^n$ | 85.2 | 64.3 | 73.3 | **71.3** | **35.0** | **47.0** | **70.8** | **7.1** | **12.9** |
| | CL | – | ✓ | $\mathbb{H}_L^n$ | **86.4** | 64.9 | 74.1 | 65.6 | 32.4 | 43.4 | 66.9 | 6.5 | 11.8 |
| | EL+CL | Pos. | ✓ | $\mathbb{H}_L^n$ | 84.7 | 63.1 | 72.3 | 63.0 | 22.8 | 33.5 | 64.2 | 6.6 | 11.9 |
| | SEL | Pos.+Neg. | ✗ | $\mathbb{H}_L^n$ | 82.7 | 65.9 | 73.4 | 62.1 | 29.2 | 39.7 | 63.1 | 6.6 | 12.0 |
| | SEL+CL | Pos.+Neg. | ✗ | $\mathbb{H}_L^n$ | 83.6 | **66.9** | **74.3** | 63.3 | 33.1 | 43.5 | 67.6 | 6.4 | 11.7 |
| | SEL+CL | Pos.+Neg. | ✓ | $\mathbb{H}_L^n$ | 85.8 | 64.8 | 73.9 | 64.8 | 27.5 | 38.6 | 64.8 | 6.2 | 11.4 |
| Species | CLIBD | – | ✓ | $\mathbb{R}^n$ | 81.8 | 60.6 | 69.7 | **55.1** | **24.3** | **33.7** | **55.8** | 0.7 | 1.4 |
| | CL | – | ✓ | $\mathbb{H}_L^n$ | **84.4** | 61.8 | **71.4** | 48.2 | 22.6 | 30.8 | 53.7 | 0.9 | 1.7 |
| | EL+CL | Pos. | ✓ | $\mathbb{H}_L^n$ | 82.5 | 60.1 | 69.6 | 45.4 | 14.3 | 21.8 | 50.5 | 0.9 | 1.8 |
| | SEL | Pos.+Neg. | ✗ | $\mathbb{H}_L^n$ | 79.5 | 62.3 | 69.9 | 45.5 | 20.0 | 27.8 | 52.0 | **1.1** | **2.1** |
| | SEL+CL | Pos.+Neg. | ✗ | $\mathbb{H}_L^n$ | 80.5 | **63.2** | 70.8 | 46.8 | 22.8 | 30.7 | 54.2 | 0.7 | 1.4 |
| | SEL+CL | Pos.+Neg. | ✓ | $\mathbb{H}_L^n$ | 82.6 | 62.0 | 70.8 | 47.8 | 19.0 | 27.2 | 51.4 | 1.0 | **2.1** |

## 4.2. Results

We compare our hyperbolic SEL strategy against baselines on the BIOSCAN-1M dataset across three retrieval tasks (DNA-to-DNA, Image-to-Image, and Image-to-DNA) evaluated at four taxonomic levels (order, family, genus, and species). We investigate how well training with contrastive loss (CL) in the hyperbolic space performs compared with training in Euclidean space (CLIBD [8]). We then compare different ways of training in hyperbolic space, comparing a strategy similar to MERU [3] with entailment loss and contrastive losses (EL + CL) to different variants of SEL. Table 3 reports macro Top-1 accuracy for seen and unseen taxa, as well as their harmonic mean.

Across all retrieval tasks, models achieve high accuracy at the coarsest levels, but this falls off substantially as ranks become more fine-grained, especially for image-based retrieval. We note that hyperbolic models consistently achieve results that are comparable to the Euclidean CLIBD baseline across all ranks and retrieval settings. SEL methods consistently perform best at unseen DNA retrieval, whereas the Euclidean model performs best at image retrieval. Comparing EL+CL to SEL+CL (both with full text), we find that SEL+CL always dominates the former, showing the utility of the stacked entailment over single-layer entailment. Comparing SEL+CL with and without full text, we find full text supervision improves unimodal seen taxa retrieval, but decreases unseen taxa and cross-modal performance.

## 5. Discussion

Our experiments demonstrate that hyperbolic learning can effectively capture hierarchical structure in biological data and provides performance competitive with established Euclidean methods. However, neither framework fully overcomes the persistent challenge of fine-grained, open-world species identification.

Improving classification at fine-grained taxonomic ranks and for novel, unseen taxa remains a key direction for future work. Potential strategies include addressing class imbalance, enhancing data augmentation, or leveraging more advanced hierarchical or uncertainty-aware methods.

## Acknowledgement

We acknowledge the support of the Government of Canada's New Frontiers in Research Fund (NFRF) [NFRFT-2020-00073]. AXC and GWT are also supported by Canada CIFAR AI Chair grants. JBH is supported by the Pioneer Centre for AI (DNRF grant number P1). This research was enabled in part by support provided by the Digital Research Alliance of Canada (alliancecan.ca) and a CFI/BCKDF JELF.

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

# Appendix

In this appendix, we provide details of our model (Appendix A) and additional results (Appendix B).

## A. Model Details

### A.1. Overall Architecture

Our framework comprises three modality-specific encoders, following the setup used in CLIBD [8]:

- **Image encoder:** We employ a ViT-B[1] backbone, initialized with ImageNet-21k pretraining and further tuned on ImageNet-1k [5].
- **DNA encoder:** BarcodeBERT [1] with 5-mer tokenization, pretrained via masked language modeling on $893\,\mathrm{k}$ DNA barcode sequences [4]. This corpus is related to but does not overlap with BIOSCAN-1M, making it suitable for unbiased evaluation.
- **Text encoder:** A pretrained BERT-Small model [20] is used to embed taxonomic labels at different ranks.

Each encoder produces Euclidean embeddings of size $d = 768$, which are mapped to a Lorentzian hyperbolic space $\mathbb{H}_L^n$ (with curvature $c > 0$) via an exponential map described in Section A.2.

---

[1]Implemented as `vit_base_patch16_224` from the `timm` library.

## A.2. Hyperbolic Projection and Distances

Following MERU [3], we project encoder outputs (Euclidean vectors) onto the Lorentzian hyperboloid using the exponential map. Here, $c > 0$ denotes the curvature of the hyperbolic space $\mathbb{H}_c^d$; smaller values of $c$ correspond to a "flatter" geometry, while larger values lead to more strongly curved spaces.

The general exponential map from a tangent vector $v \in T_p\mathbb{H}_c^d$, where $T_p\mathbb{H}_c^d$ denotes the tangent space at point $p$ in the Lorentz model of $\mathbb{H}_c^d$, to the manifold is given by:

$$\exp_p(v) = \cosh\left(\sqrt{c}\,\|v\|_{\mathbb{L}}\right) p + \frac{\sinh\left(\sqrt{c}\,\|v\|_{\mathbb{L}}\right)}{\sqrt{c}\,\|v\|_{\mathbb{L}}}\, v.$$

Hyperbolic distances are computed via:

$$d_{\mathbb{H}}(x, y) = \frac{1}{\sqrt{c}} \cosh^{-1}\left(-\langle x, y\rangle_{\mathbb{L}}\right).$$

## A.3. Entailment Cones

The half-aperture angle of the cone centred at $u$ is:

$$\alpha(u) = \sin^{-1}\left(\frac{K}{\|u\|_{\mathbb{H}}}\right),$$

where $K = 2r_{\min}/\sqrt{c}$. Here $r_{\min}$ is a small constant $0.1$, which is used to set boundary conditions near the origin and prevent $\alpha(u)$ from diverging when $\|u\|_{\mathbb{E}}$ is small.

## A.4. Input Text Construction

Taxonomic labels are encoded per rank using the BERT-Small tokenizer [20]. Full-text inputs concatenate all ranks with spaces as separators (see Table 2).

Table 2. Example of taxonomic labels and their full-text concatenation.

| Rank | Label |
|---|---|
| Order | Hymenoptera |
| Family | Formicidae |
| Genus | Myrmica |
| Species | Myrmica specioides |
| Full-text | Hymenoptera Formicidae Myrmica Myrmica specioides |

## A.5. Training Details

We train on the `train_seen` split (36k samples) of BIOSCAN-1M. The batch size is 2000 for CL-only runs and 1520 for SEL runs across $4\times$A100 (80GB). Optimization is with Adam ($\beta_1 = 0.9$, $\beta_2 = 0.98$, weight decay $1\mathrm{e}{-4}$), with a one-cycle LR schedule ($1\mathrm{e}{-6}$ to $5\mathrm{e}{-5}$). Mixed precision is used. All negatives come from in-batch sampling; for entailment loss, negatives are taxonomy-aware.

## B. Additional results

In the main paper, we reported the macro averaged accuracy over classes for the different methods. Here in Table 3

Table 3. Micro top-1 accuracy (%) comparison of different training objectives across taxonomic levels on BIOSCAN-1M. CL: contrastive loss. EL: entailment loss; SEL: stacked entailment loss; We evaluate uni- and multi-modal retrieval tasks including DNA-to-DNA, Image-to-Image, and Image-to-DNA. Accuracy is reported on both seen and unseen taxa, along with their harmonic mean (H.M.). Each method is further characterized by the configuration of entailment loss used (EL config.), whether full taxonomic text embedding is included utilized during training (Full Text), and the choice of embedding space (Euclidean: $\mathbb{R}^n$, or Lorentzian-hyperbolic: $\mathbb{H}_L^n$). All models are trained on the `train_seen` split of CLIBD and evaluated on the `test` split. **Best** results are shown in bold; second-best are underlined.

| Rank | Method | EL Settings | Full Text | Space | DNA-to-DNA | | | Image-to-Image | | | Image-to-DNA | | |
|---|---|---|---|---|---|---|---|---|---|---|---|---|---|
| | | | | | Seen | Unseen | H.M. | Seen | Unseen | H.M. | Seen | Unseen | H.M. |
| Order | CLIBD | - | ✓ | $\mathbb{R}^n$ | **99.2** | 98.2 | **98.7** | **99.6** | **98.3** | **98.9** | 99.4 | **96.4** | **97.9** |
| | CL | - | ✓ | $\mathbb{H}_L^n$ | 99.1 | 98.0 | 98.6 | 99.4 | 98.0 | 98.7 | **99.5** | 95.5 | 97.5 |
| | EL+CL | Pos. | ✓ | $\mathbb{H}_L^n$ | **99.2** | 97.9 | 98.6 | 99.3 | 97.1 | 98.2 | 99.2 | 95.8 | 97.4 |
| | SEL | Pos.+Neg. | ✗ | $\mathbb{H}_L^n$ | 99.1 | 98.2 | 98.6 | 99.4 | 97.7 | 98.6 | 99.1 | 95.0 | 97.0 |
| | SEL+CL | Pos.+Neg. | ✗ | $\mathbb{H}_L^n$ | 99.1 | **98.3** | **98.7** | 99.4 | 97.7 | 98.5 | 98.9 | 95.5 | 97.2 |
| | SEL+CL | Pos.+Neg. | ✓ | $\mathbb{H}_L^n$ | **99.2** | 98.1 | 98.6 | 99.4 | 97.9 | 98.6 | 99.1 | 96.0 | 97.5 |
| Family | CLIBD | - | ✓ | $\mathbb{R}^n$ | **97.5** | 91.8 | 94.6 | **95.4** | **85.7** | **90.3** | **94.8** | **69.7** | **80.4** |
| | CL | - | ✓ | $\mathbb{H}_L^n$ | 97.1 | 91.8 | 94.4 | 94.3 | 84.7 | 89.2 | 93.9 | 68.1 | 79.0 |
| | EL+CL | Pos. | ✓ | $\mathbb{H}_L^n$ | 97.2 | 90.6 | 93.8 | 93.5 | 80.3 | 86.4 | 93.2 | 66.4 | 77.6 |
| | SEL | Pos.+Neg. | ✗ | $\mathbb{H}_L^n$ | 97.0 | **92.5** | **94.7** | 93.4 | 83.0 | 87.9 | 92.5 | 67.2 | 77.8 |
| | SEL+CL | Pos.+Neg. | ✗ | $\mathbb{H}_L^n$ | 96.7 | 92.4 | 94.5 | 93.6 | 83.9 | 88.5 | 93.0 | 67.5 | 78.2 |
| | SEL+CL | Pos.+Neg. | ✓ | $\mathbb{H}_L^n$ | 97.1 | 91.3 | 94.1 | 94.3 | 83.3 | 88.5 | 93.8 | 68.6 | 79.2 |
| Genus | CLIBD | - | ✓ | $\mathbb{R}^n$ | 94.8 | 85.1 | 89.7 | **88.2** | **69.0** | **77.4** | **87.1** | **37.1** | **52.1** |
| | CL | - | ✓ | $\mathbb{H}_L^n$ | **95.3** | 85.6 | 90.2 | 85.6 | 68.0 | 75.8 | 86.0 | 36.1 | 50.8 |
| | EL+CL | Pos. | ✓ | $\mathbb{H}_L^n$ | 95.1 | 84.6 | 89.5 | 83.2 | 60.4 | 70.0 | 84.5 | 34.6 | 49.1 |
| | SEL | Pos.+Neg. | ✗ | $\mathbb{H}_L^n$ | 94.0 | 86.1 | 89.9 | 83.7 | 65.7 | 73.6 | 83.2 | 35.3 | 49.5 |
| | SEL+CL | Pos.+Neg. | ✗ | $\mathbb{H}_L^n$ | 94.2 | **86.7** | **90.3** | 83.8 | 67.0 | 74.5 | 84.4 | 34.1 | 48.6 |
| | SEL+CL | Pos.+Neg. | ✓ | $\mathbb{H}_L^n$ | 95.0 | 85.5 | 90.0 | 84.8 | 65.3 | 73.8 | 84.2 | 35.0 | 49.4 |
| Species | CLIBD | - | ✓ | $\mathbb{R}^n$ | 93.0 | 82.0 | 87.2 | **77.4** | **53.4** | **63.2** | **78.3** | 1.9 | 3.6 |
| | CL | - | ✓ | $\mathbb{H}_L^n$ | **93.6** | 82.7 | **87.8** | 73.3 | 52.1 | 60.9 | 77.6 | **2.4** | **4.6** |
| | EL+CL | Pos. | ✓ | $\mathbb{H}_L^n$ | 93.5 | 81.6 | 87.2 | 69.8 | 44.5 | 54.4 | 75.9 | 1.5 | 2.9 |
| | SEL | Pos.+Neg. | ✗ | $\mathbb{H}_L^n$ | 91.8 | 83.2 | 87.3 | 71.8 | 50.2 | 59.1 | 75.2 | 1.4 | 2.8 |
| | SEL+CL | Pos.+Neg. | ✗ | $\mathbb{H}_L^n$ | 92.1 | **83.5** | 87.6 | 71.8 | 51.2 | 59.8 | 75.7 | 1.6 | 3.1 |
| | SEL+CL | Pos.+Neg. | ✓ | $\mathbb{H}_L^n$ | 93.3 | 82.7 | 87.7 | 72.7 | 49.4 | 58.8 | 75.6 | 2.0 | 3.8 |

we report the micro accuracy, averaging over individual instances. Compared to the macro accuracy, which treat all classes evenly, the micro accuracy will give more weight to classes with more instances. Overall, we see a similar trends in the comparative performance of the different methods for both macro and micro averaged results, with the micro averaged accuracy being substantially higher (as the macro averaged accuracy is pulled down by rare classes).

