# OpenReview forum: "Hyperbolic Multimodal Representation Learning for Biological Taxonomies"
_thecvf.com/ICCV/2025/Workshop/BEW — BEW 2025 Abstract_

### Official Review · Reviewer_5quq · 2025-06-30
**An interesting paper for the workshop**

**Rating:** 4
**Confidence:** 4

**Review:**

The paper investigates the embedding space of hyperbolic networks using contrastive learning and entailment-based objectives.  The experiments are done on multi-modal inputs on the BIOSCAN-1M dataset (images with paired DNA).

Strengths:
- The paper compares a version of previous work (MERU, EL+CL) and different variants of stacked entailment loss in the multi- and uni-modal setups
- The results show the contributions of different configurations at different levels in the semantic hierarchy

Weaknesses:
- The results show partial improvements
- The taxonomy classification performances in the structure could better displayed (e.g., include plots for the embeddings and their characteristics)

Despite the shortcomings, the ideas presented in the paper are interesting.  The paper could spark interesting discussions during the workshop.  Thus, despite the limitations, I recommend its acceptance.

---

### Official Review · Reviewer_DKHy · 2025-07-04

**Rating:** 4
**Confidence:** 4

**Review:**

Firstly, there seems to be a problem with the template as the first pape is odd.

In terms of the science, it presents a hyperbolic multimodal learning framework for classifying biological specimens using multiple data types, i.e. images, DNA barcodes, and taxonomic labels. The central idea is to embed all modalities into a shared hyperbolic space, better capturing the hierarchical structure inherent in biological taxonomies.

The proposed method extends prior multimodal contrastive approaches by introducing a stacked entailment loss (SEL) that geometrically enforces parent-child relationships across taxonomic levels in hyperbolic space. Modality-specific encoders produce embeddings that are projected into Lorentzian hyperbolic space, enabling both intra- and inter-modal alignment with taxonomic hierarchy.

Experiments on the BIOSCAN-1M dataset show that the hyperbolic models:

--Perform competitively with or better than Euclidean baselines at higher taxonomic ranks (order, family).

--Improve retrieval and classification of unseen species when using DNA inputs.

--Struggle—like all current models—with fine-grained species-level classification, especially from image-only inputs.

The stacked entailment loss outperforms simpler entailment or contrastive-only baselines, particularly when combined with full taxonomic text supervision. However, performance trade-offs are observed between seen and unseen taxa depending on the loss formulation.

Please make sure you fix the template.

---

### Decision · Program_Chairs · 2025-07-09

**Decision:**

Accept (Abstract)

**Comment:**

The reviewers agree that this paper will be an interesting contribution for the poster discussion during the workshop.